# VIRDO++: Real-World, Visuo-Tactile Dynamics and Perception of Deformable Objects

**Youngsun Wi**[1]    **Andy Zeng**[2]    **Pete Florence**[2]    **Nima Fazeli**[1]
[1]Robotics Department, University of Michigan    [2]Robotics at Google
{yswi, nfz}@umich.edu    {andyzeng, peteflorence}@google.coms
https://www.mmintlab.com/virdopp

**Abstract:** Deformable objects manipulation can benefit from representations that seamlessly integrate vision and touch while handling occlusions. In this work, we present a novel approach for, and real-world demonstration of, multimodal visuo-tactile state-estimation and dynamics prediction for deformable objects. Our approach, VIRDO++, builds on recent progress in multimodal neural implicit representations for deformable object state-estimation [1] via a new formulation for deformation dynamics and a complementary state-estimation algorithm that (i) maintains a belief distribution of deformation within a trajectory, and (ii) enables practical real-world application by removing the need for contact patches. In the context of two real-world robotic tasks, we show: (i) high-fidelity cross-modal state-estimation and prediction of deformable objects from partial visuo-tactile feedback, and (ii) generalization to unseen objects and contact formations.

**Keywords:** Deformable Object Manipulation, Multimodal Representation Learning

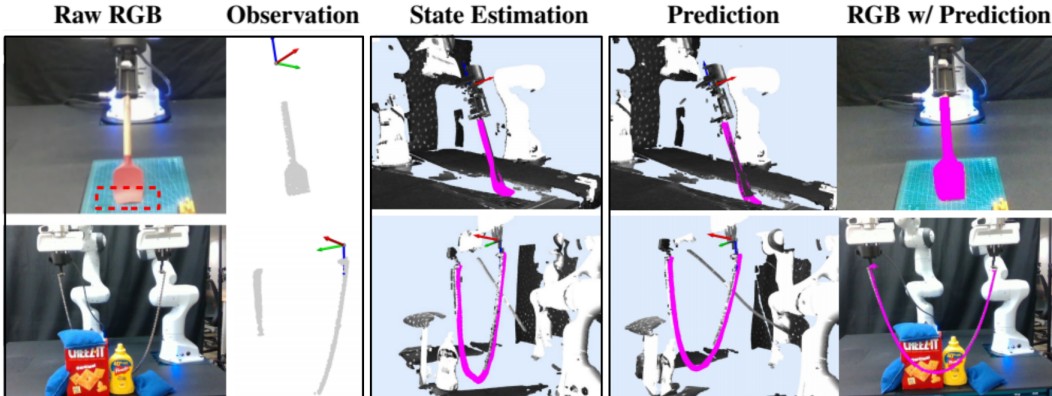

Fig. 1: Given partial point cloud observations (left), and multimodal sensor measurements (end effector wrist reaction wrench and pose), VIRDO++ can accurately predict the 3D geometry (magenta) and reaction forces of deformable objects such as spatulas (top) and bike chains (bottom) conditioned on the next robot action under severe occlusions – which can be artificial (red dotted box) or natural due to obstacles (e.g., from other objects).

## 1   Introduction

Deformable objects are ubiquitous in our everyday lives – the clothes we wear, the food we eat, and many of the tools we use are just a few examples. As such, helpful robots of the future may benefit from the ability to master dexterous manipulation of deformable objects. At the heart of this mastery is the interplay between object geometry and force transmission, perceived by vision and touch. Deformable object manipulation is difficult due to the complexity of this interplay (e.g., infinite dimensional state-spaces and nonlinearity) and the ensuing challenges in perception (e.g., partial observability and occlusion) and controls [2, 3, 4, 5]. Multimodal visuo-tactile representations can help address many of these challenges by exploiting mutual information and complementary cues.

6th Conference on Robot Learning (CoRL 2022), Auckland, New Zealand.

Existing deformable object manipulation approaches typically use one modality (mostly vision) and rely on finite element/particle-based techniques [6, 7, 8, 9, 10, 11, 12, 13] or leverage deep learning for visual affordance/latent dynamics learning [14, 15, 16, 17, 18, 19]. The former methods typically rely on privileged knowledge (e.g., occluded or unknown boundary conditions) and stop at system identification, limiting their ability to refine the underlying physics model by learning from data. Latter methods lack (i) tactile feedback which is far more informative of contact than vision, and (ii) structured representations which can limit generalization and increase sample-complexity.

Here, we build on recent work [1] that provides a framework for visuo-tactile state estimation using multimodal implicit neural representations. This approach offers a number of advantages including multimodal sensory fusion, direct integration with raw sensory feedback, and computational speed. However, [1] does not address (i) deformation predictions (dynamics) and (ii) is not directly applicable to real-world robotics settings. The first limitation is due to the lack of a dynamics module and an emphasis on static scenes. The second limitation is due to the need for the contact patch which are typically impractical to measure in real-world robotic settings.

The primary contribution of this work is (i) a novel approach to, and (ii) real-world demonstration of, performing multi-modal visuo-tactile state estimation and dynamics prediction for deformable objects. Fig. 1 shows how our proposed approach uses partial views and tactile feedback for deformation prediction and state-estimation in real-world applications. Our contributions are:

**1. Representation of deformable object *dynamics* conditioned on actions:** Multimodal neural implicit representation with action-conditioned dynamics to predict future deformations. To demonstrate the utility of the dynamics model, we introduce a state estimation algorithm based on particle filtering to maintain a belief distribution over object deformations.

**2. Real-world demonstrations on two challenging tasks:** Scraping with a spatula and chain manipulation. We demonstrate both 3D geometry estimation in heavy occlusion and action-conditioned deformation/reaction wrench prediction. This is enabled by introducing a novel contact latent vector that eliminates the need for explicit contact patch information used in [1]. We use these tasks to demonstrate **generalization to unseen objects and novel environments**.

## 2   Related Work

**Deformable object modeling.** Recent studies have attempted to address challenges in system identification and high computation costs in conventional continuum mechanics [6, 7, 8, 9, 10, 11, 12, 13]. To ease the burden of system identification, prior studies infer the physics models' parameters based on high-fidelity physics engines [20, 21, 22] and simple force-deformation relationship (e.g. Hooke's law) [23]. The inherent limitations of these approaches are (i) that performance is confined to the underlying physics model, often with strong assumptions on the objects (uniform density/elasticity) and the force-deformation relationship (linearity), and (ii) relies on access to privileged information such as occluded or unknown boundary conditions. More recent approaches propose computationally efficient elastic object modeling with minimum system identification using, for example, a potential energy propagation [24] and geometric motion estimates [25]. However, these methods suffer from balancing between model approximation and computation cost as well as maintaining seamless integration with robotic sensing modalities (RGB-D cameras and F/T sensors). In this study, we adopt a data-driven approach using neural networks [1, 26, 27]. Learning directly from observation, our approach does not require object parameters (Young's modulus, Poisson's ratio) or a strong assumption of object composition. Even so, it can capture complex non-linear elastic behaviors and exhibit fast computation times due to the architecture of the neural networks.

**3D geometric representation of deformable objects.** Most recent studies have adopted discrete geometry representation for deformable objects with finite resolution such as meshes [7, 10, 28] and key points [15, 21], sometimes accompanied with graph neural networks [29, 27]. These approaches have been used largely for 1D and 2D objects (rope, cloth, top view of cylindrical objects). Among them, structured representations (e.g., Mesh or GNN) are advantageous at tracking connectivity, mainly used for deformable objects with large self-occlusions (e.g. folded cloth or knot) [30, 31]. On the other hand, we address general 3D geometry representations which are not confined to specific types of objects and can work with large occlusions (e.g., wok-scraping). Our work builds on dense implicit geometric representation which include signed distances fields [1, 26, 32], occupancy [33], volume density [34], and dense descriptors [35]. These approaches have several advantages over their discrete counterparts in dealing with large occlusion, high resolution surface representation

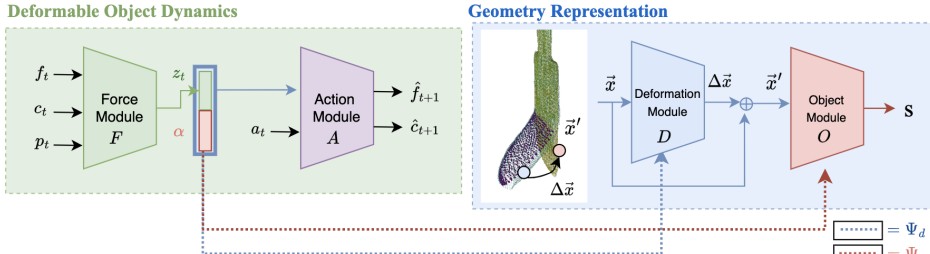

Fig. 2: **Overview.** VIRDO++ is composed of an implicit signed-distance field representation of geometry (left), informed by a deformation dynamics model (right), where dotted lines indicate hyper networks that decode embeddings into network weights.

[36], parameterization of 3D geometries [37, 38], and are useful for downstream tasks (e.g., contact location detection similar to Sec. 4.5). We extend these representations to real-world multimodal deformable object dynamics and perception by integrating touch and robot actions.

## 3 Methodology

Our approach, shown schematically in Fig. 2, is composed of a latent deformation dynamics model and an implicit dense geometric representation. Using Hidden Markov Models as an analogy, the former component plays the role of the hidden state transition, and the latter is the observation model. We expand on the latent deformation dynamics in Sec. 3.1 and discuss the implicit geometric representation in Sec. 3.2.

### 3.1 Deformable Object Dynamics

VIRDO++ represents deformable object states using latent object ($\boldsymbol{\alpha}$) and force ($\boldsymbol{z_t}$) codes. The object code condenses the undeformed shape information into a feature representation that can interpolate to unseen object variants. The force code is generated by the Force Module ($\mathbf{F}$) which encodes boundary conditions as $\boldsymbol{z_t} = \mathbf{F}(\boldsymbol{f_t}, \boldsymbol{c_t}, \boldsymbol{p_t})$ where $\boldsymbol{f_t} \in \mathbb{R}^6$ is the reaction wrench at the wrist, $\boldsymbol{p_t} \in \mathbb{R}^6$ is robot end-effector pose, and $\boldsymbol{c_t} \in \mathbb{R}^{lc}$ is the contact latent vector, where $lc$ is the dimension of the contact latent vector. The contact latent vector $\boldsymbol{c_t}$ is a learnable latent vector that can change over time whose primary role is to store contact information. However, it can also implicitly integrate information about the object's physical properties (e.g., stiffness) and historical information about the object's states and deformations. This embedding generalizes the original VIRDO [1] in terms of both (i) no longer needing known contact patch locations, unlocking the capability to run VIRDO in the real-world, and (ii) representing additional useful information beyond just the contact location as was the case in VIRDO. In more detail, VIRDO [1] uses ground truth contact patches encoded by a PointNet encoder [39] directly which required access to this privileged information. Here, the contact patch and encoder are replaced by the learnable contact latent vector.

The Action Module ($\mathbf{A}$) predicts the next step's boundary conditions given the current (latent) state and robot action: $\mathbf{A}(\boldsymbol{\alpha}, \boldsymbol{z_t}, \boldsymbol{a_t}) = \hat{\boldsymbol{f}}_{t+1}, \hat{\boldsymbol{c}}_{t+1}$ where $\hat{\boldsymbol{f}}_{t+1} \in \mathbb{R}^6$ is the predicted reaction wrench at the wrist, $\hat{\boldsymbol{c}}_{t+1}$ is the predicted contact latent vector, and $\boldsymbol{a_t} \in \mathbb{R}^6$ is the robot action (Cartesian displacement). This predictive capability enables state-estimation and downstream tasks such as contact location detection crucial for deformable object manipulation. Though we did not include in this paper, the predictive ability of VIRDO++ also enables planning for deformable object manipulation. VIRDO lacked this functionality and could only infer the object geometry statically.

### 3.2 Geometry Representation

VIRDO++ decouples the 3D geometric representation of deformable objects into an undeformed (nominal) shape signed-distance field (SDF) and a set of deformation fields, similar to [1, 38]. The deformation field is a 3D vector field that, when summed with the deformed object SDF, results in the nominal shape SDF. More precisely, any query point $\boldsymbol{x} \in \mathbb{R}^3$ in the robot wrist frame belonging to the deformed shape SDF can be mapped back to the corresponding point $\boldsymbol{x}'$ of the nominal shape SDF by applying the point-wise deformation field $\Delta\boldsymbol{x}$ as $\boldsymbol{x} + \Delta\boldsymbol{x} = \boldsymbol{x}'$.

**Nominal Shape Representation:** The Object Module $\mathbf{O}(\boldsymbol{x}')$ is the parametric representation of the nominal shape SDF – the object shape with no external contacts corresponding to $\Delta\boldsymbol{x} = 0$. Here, we use a feedforward neural network to parameterize the SDF [37] and write $\mathbf{O}(\boldsymbol{x}'|\boldsymbol{\Psi}_o(\boldsymbol{\alpha})) = s$ where $s$ is the signed-distance at query point $\boldsymbol{x}'$ and $\boldsymbol{\Psi}_o(\boldsymbol{\alpha})$ is a hyper-network that predicts the weights

of $\mathbf{O}$ conditioned on the object code $\boldsymbol{\alpha}$. The object code and hyper-network weights are learned end-to-end in an auto-decoder approach [37]. We simplify our notation to $\mathbf{O}(\boldsymbol{x}')$ for the remainder.

**Deformation Field Representation:** The generalized geometric representation of a deformable object is given by $\text{SDF}(\boldsymbol{x}) = \mathbf{O}_{\boldsymbol{\Phi}_o}\big(\boldsymbol{x} + \mathbf{D}(\boldsymbol{x}|\Psi_d(\boldsymbol{z}_t, \boldsymbol{\alpha}_t))\big) = s$. Here, the Deformation Module $\mathbf{D}(\boldsymbol{x}|\Psi_d(\boldsymbol{z}_t, \boldsymbol{\alpha}_t)) = \Delta\boldsymbol{x}$ produces an object's deformation field given force and object code pairs. Similar to the Object Module, Deformation Module's weights are given by the hyper-network $\Psi_d$.

## 3.3 Training and Loss Formulation

The training dataset for VIRDO++ is a set of trajectories $\mathcal{S} = \{\mathcal{T}_1, \mathcal{T}_2, ..., \mathcal{T}_N\}$, where each trajectory $\mathcal{T}_j = \{(\boldsymbol{p}_0, \boldsymbol{f}_0, \boldsymbol{P}_0, \boldsymbol{a}_0), (\boldsymbol{p}_1, \boldsymbol{f}_1, \boldsymbol{P}_1, \boldsymbol{a}_1), ..., (\boldsymbol{p}_k, \boldsymbol{f}_k, \boldsymbol{P}_k, \boldsymbol{a}_k)\}$ has $k$ sequential observation and action tuples. Here $\boldsymbol{P}_j$ is a partial point cloud. Our first step is to pretrain a nominal shape representation (i.e., hyper-network $\boldsymbol{\Psi_o}$ and object code $\boldsymbol{\alpha}$). The loss for training $m$ nominal shapes is $\mathcal{L}_{nominal} = \mathcal{L}_{sdf} + \lambda_1 \mathcal{L}_{latent} + \lambda_2 \mathcal{L}_{hyper}$ as in [1], where $\mathcal{L}_{latent} = \sum_{i=1}^{m} \|\boldsymbol{\alpha}^i\|_2$, $\mathcal{L}_{hyper} = \sum_{i=1}^{m} \|\boldsymbol{\Psi}_o(\boldsymbol{\alpha}^i)\|_2$, and:

$$\mathcal{L}_{sdf} = \sum_{i=1}^{m} \Big( \sum_{\bar{\boldsymbol{x}} \in \boldsymbol{\Omega}} |clamp(\mathbf{O}^i(\boldsymbol{x}), \delta) - clamp(s^*, \delta)| + \lambda \sum_{\bar{\boldsymbol{x}} \in \boldsymbol{\Omega}_0} (1 - \langle \nabla \mathbf{O}^i(\boldsymbol{x}), \boldsymbol{n}^* \rangle) \Big).$$

$\boldsymbol{n}^*$ is the ground truth normal vector, $\delta$ is a parameter that clips SDF predictions, and $s^*$ is the ground truth signed distance.

After the pretraining, we train the rest of the modules using the three losses $\mathcal{L}^{geo}, \mathcal{L}^{pred}$ and $\mathcal{L}^{reg}$ summed over a fixed horizon $(w)$ while freezing $\Psi_o$ and the object code $\alpha$:

$$\mathcal{L}_{tot} = \sum_{t=t_0}^{t_0+w-1} (\mathcal{L}_t^{geo} + \lambda_3 \mathcal{L}_t^{pred} + \lambda_4 \mathcal{L}_t^{reg}). \tag{1}$$

The geometry representation loss is defined as:

$$\mathcal{L}^{geo} = \lambda_5 \underbrace{\sum_{\boldsymbol{x} \in \boldsymbol{\Omega}} \|\mathbf{D}_{\boldsymbol{\Psi}_d}(\boldsymbol{x})\|_2}_{\text{minimum correction}} + \lambda_6 \underbrace{\text{CD}(\boldsymbol{P} + \mathbf{D}_{\boldsymbol{\Psi}_d}(\boldsymbol{P}), \bar{\boldsymbol{P}}^*)}_{\text{correspondence}} + \lambda_7 \underbrace{\sum_{\boldsymbol{x} \in \boldsymbol{\Omega}_0} (1 - \langle \nabla_{\boldsymbol{x}} \mathbf{O}_{\boldsymbol{\Psi}_o}(\boldsymbol{x}'), \boldsymbol{n}^* \rangle}_{\text{normal aligning}}$$
$$+ \lambda_8 \underbrace{\sum_{\boldsymbol{x} \in \boldsymbol{\Omega}} |clip(\mathbf{O}_{\boldsymbol{\Psi}_o}\big(\boldsymbol{x} + \mathbf{D}_{\boldsymbol{\Psi}_d}(\boldsymbol{x}), \delta\big) - clip(s^*, \delta)|}_{\text{signed distance regression}}. \tag{2}$$

where $\boldsymbol{x}' = \boldsymbol{x} + \mathbf{D}_{\boldsymbol{\Psi}_d}(\boldsymbol{x})$, $\Omega$ is the 3D querying space, $\Omega_0 \subset \Omega$ is the on-surface region, $\mathbf{P} := \{\boldsymbol{p}|\boldsymbol{p} \in \Omega_0\}$ is an unordered set of the on-surface points, and $\bar{\boldsymbol{P}}^*$ is the ground truth nominal shape point cloud. Next, we add a reaction wrench prediction loss as $\mathcal{L}_{pred} = \|\boldsymbol{f}_{t+1} - \hat{\boldsymbol{f}}_{t+1}\|$. The prediction loss for $\hat{\boldsymbol{c}}_{t+1}$ is implicitly handled by Eq. 2 because the contact latent vector prediction is recursively used as next step's input during training. In contrast, we do replace $\hat{\boldsymbol{f}}_{t+1}$ with real measurement in Eq. 2. Finally, regularization losses $\mathcal{L}_t^{reg} = \lambda_9 \|\boldsymbol{z}_t\|_2 + \lambda_{10} \|\boldsymbol{c}_t\|_2 + \lambda_{11} \|\boldsymbol{\Psi}_d(\boldsymbol{\alpha}, \boldsymbol{z}_t)\|_2$, which regularize the force codes, contact latent vector, and the weights of $\mathbf{D}_{\boldsymbol{\Psi}_d}(\boldsymbol{x})$ respectively.

## 3.4 Inference and State-estimation Algorithm

To maintain a belief distribution over the object deformation, we develop a particle filter-based inference algorithm, Alg. 1. Each particle represents a contact latent vector which can be used to reconstruct the full 3D object geometry. The `Refine` step updates the particle set by performing gradient descent on $\min_{\hat{\mathbf{C}}_t} \mathcal{L}^{geo}(\boldsymbol{P}_t, \hat{\mathbf{C}}_t, \boldsymbol{f}_t, \boldsymbol{p}_t)$ with the visible object point cloud. The particles are then weighted according to their wrench error prediction with `Weight-Function` defined as $\boldsymbol{w}_t^i = exp(-\gamma(\hat{\boldsymbol{f}}_t^i - \boldsymbol{f}_t))$ and resampled with probability proportional to this weight vector. Here, $\hat{\boldsymbol{f}}_t^i$ is the reaction wrench prediction of the $i^{th}$ particle and $\boldsymbol{f_t}$ is a ground truth measurement. Finally, the particles are propagated through the Force and Action Modules, a small amount of uncertainties is applied $\sim \mathcal{N}(0, 0.01)$ to the particles and the algorithm is repeated. (see algorithm variations in **Appendix** Sec. A.2.2)

---

**Algorithm 1** Particle Filter based Inference Algorithm

---

1: **procedure**
2:     $\hat{\mathbf{C}}_0 \leftarrow \{\hat{\boldsymbol{c}}_0^0, \hat{\boldsymbol{c}}_0^1, ..., \hat{\boldsymbol{c}}_0^{n-1}\} \sim \mathcal{N}(0, 0.01)$                                             ▷ Init Particles
3:     **for** each $scan$ in $Trajectory\ Length$ **do**
4:         $\mathbf{C}_t \leftarrow \texttt{Refine}(\mathbf{P}_t, \hat{\mathbf{C}}_t, \boldsymbol{f}_t, \boldsymbol{p}_t)$
5:         $\boldsymbol{w}_t \leftarrow \texttt{Weight-Function}(\boldsymbol{f}_t, \hat{\boldsymbol{f}}_t^0, \hat{\boldsymbol{f}}_t^1, ..., \hat{\boldsymbol{f}}_t^{n-1})$
6:         $\bar{\mathbf{C}}_\mathbf{t} \leftarrow \texttt{Re-sampling}(\boldsymbol{w}_t, \mathbf{C}_t)$
7:         $\mathbf{Z}_\mathbf{t} \leftarrow \texttt{Force-Module}(\bar{\mathbf{C}}_\mathbf{t}, \boldsymbol{f}_t, \boldsymbol{p}_t, \alpha)$
8:         $\{(\hat{\boldsymbol{f}}_{t+1}^0, \hat{\boldsymbol{c}}_{t+1}^0), (\hat{\boldsymbol{f}}_{t+1}^1, \hat{\boldsymbol{c}}_{t+1}^1), ..., (\hat{\mathbf{f}}_{t+1}^{n-1}, \hat{\boldsymbol{c}}_{t+1}^{n-1})\} \leftarrow \texttt{Action-Module}\ (\mathbf{Z}_\mathbf{t}, \boldsymbol{\alpha}, \boldsymbol{a}_t)$
9:         $t \leftarrow t + 1$
10:    **end for**
11: **end procedure**

---

## 4 Experiments

For real robot experiments, we investigate two classes of objects: **spatulas** and **bike chains**. These objects exhibit different deformation behaviors: (i) spatulas are representative of bendable elastic objects that retain their shape when external forces are removed, whereas (ii) bike chains (with the 2 end points controlled by a bi-manual manipulator) can undergo significantly larger deformations (e.g., from gravity) – requiring algorithms that can reason over free-space while estimating in-contact shapes and forces. Spatula deformations are generally local and experience significant reaction wrenches, while chain deformations are much larger but with much smaller reaction wrenches.

### 4.1 Spatula Manipulation Dataset

Train and test objects are shown in Fig. 3. The training set consists of 4 spatulas, each with 672 samples ($N = 28$, $k = 24$), using 9:1 train/test split (25 train and 3 test trajectories) for representation learning. The test dataset consists of 2 held-out spatulas, each with 72 samples ($N = 3$, $k = 24$), and is used to evaluate zero-shot transfer and not used for representation learning. At the beginning of each trajectory ($t = 0$), the end-effector moves to a random world Cartesian position in a bounding box of size $(0.1 \times 0.24 \times 0.06)$ meters and the tool is brought into contact with the table. Within a trajectory, the tool maintains contact while sampling a random action from $(x, y, z, r, p, y) = \pm(0.02, 0.02, 0.01, 0.06, 0.025, 0.04)$ where translations are in meters and rotations in radians. Wrench measurements are averaged over 0.05 [s] and the point cloud is registered to the wrist frame. The method

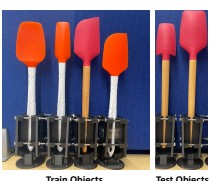

**Train Objects**     **Test Objects**

Fig. 3: Train and test objects differ in terms of (i) geometry, and (ii) how they react under wrench.

only requires a full point cloud from the nominal geometry of the object. This is collected by holding the object in front of the camera and rotating. Deformed object geometries are collected from a single camera viewing the front side of the spatulas without occlusions and are all partial. $\boldsymbol{P}$ is then the result of segmenting the object pointcloud and normalizing the scale into a bounding box of size $2 \times 2 \times 2 [m]$ centered around $(0, 0, 0)$. The test dataset point clouds $\boldsymbol{P}$ contain occlusions at the bottom $0.15 \times$ `spatula-height` as shown in the top observation panel in Fig. 1. For hardware, we use Franka Emika's Panda, wrist mounted ATI Gamma F/T sensor, and Photoneo PhoXi 3D L.

### 4.2 State Estimation and Predictions

Here, we evaluate the deformation prediction (Sec. 3.1) and state-estimation (Alg. 1) accuracy of VIRDO++ on training and test trajectories. Fig. 4 shows an example of state-estimation for a test trajectory where the ground truth (black) and reconstruction (magenta) are overlaid. We note the high state-estimation accuracy indicated by the overlay agreement, where the average distance between the measured points and reconstructions is within $0.03[m]$ of normalized scale, just 1% of the object length.

| CD ($\times 10^3$) | Spatula w/ table | Bike Chain |
|---|---|---|
| Train Est. | 0.777 (0.215) | 0.708 (0.331) |
| Train Pred. | 0.830 (0.330) | 0.762 (0.224) |
| Test Est. | 0.934 (0.210) | 0.760 (0.234) |
| Test Pred. | 1.041 (0.348) | 1.247 (0.391) |

Table. 1: VIRDO++ can accurately estimate object geometry deformations under significant occlusions, measured in terms of mean Chamfer distances (CD) (std.) $[m^2]$ scaled as ($\times 10^3$) across 5600 on-surface points between predicted and ground truth reconstructions.

The quantitative results, Tab. 1, show VIRDO++ is able to faithfully predict and estimate object deformations and reaction forces despite significant occlusions. Force and torque norm errors are less than 0.567 [N], 0.106 [Nm] (train) and 1.130 [N], 0.135 [Nm] (test) respectively (see **Appendix** Sec. A.3.3 and Sec. A.3.4 for details).

Our ablation study, Tab. 3, helps further anchor the scale of these state-estimation and dynamics prediction errors. On state estimation, for example, our full VIRDO++ method achieves a Chamfer distance ($\times 10^3$) of normalized geometry as 0.93 $[m^2]$. The baseline model without a novel contact latent vector input has 27% increased chamfer distance error as 1.18 $[m^2]$ for geometry estimation. Without $c_t$, the baseline does not reason about contact during inference, resulting in larger errors during state-estimation. Further if we assume the object is rigid and reduce our state estimation to a "rigid pose estimation" pipeline, the Chamfer distance ($\times 10^3$) increases considerably to 16.19 $[m^2]$.

The geometry predictions are slightly less accurate than state-estimation because the geometry predictions are computed by the learned dynamics using previous observation and action, while state-estimations use the current measurement to update the prediction. Despite this, the prediction Chamfer distances are within 14% of the state estimation results. Moreover, the train and test trajectories results show 25% difference in Chamfer distance. This is interesting because the test trajectories are not only from unseen tool configurations but also with occlusions (see **Dataset**). This suggests that learned dynamics of VIRDO++ generalize effectively to unseen contact formations with occlusions.

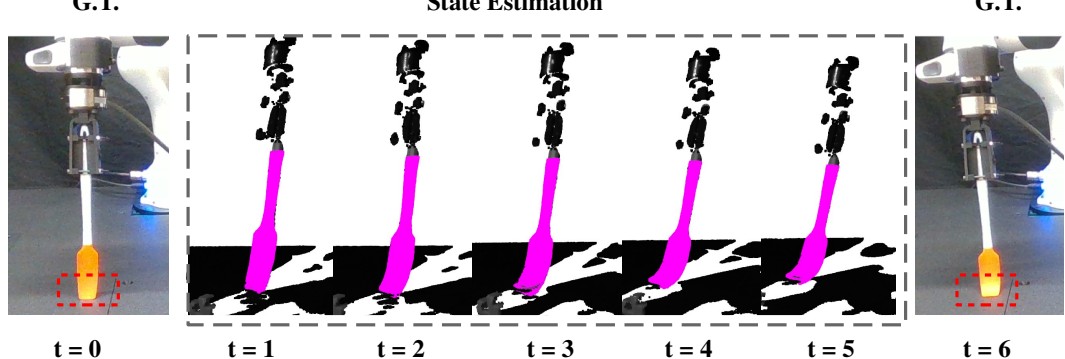

Fig. 4: Even under occlusions (generated by cropping out data within the red dotted box), our approach can accurately infer the how the geometry of a test spatula deforms over time (t = 0 to 6), indicated by high overlap agreement between the prediction (magenta) overlaid on the ground truth (black).

### 4.3   Evaluation on a Downstream Manipulation Task - Extrinsic Contact Detection

Central to compliant tool manipulation is estimating and controlling the contact between the tool and environment. For example, when scraping a wok with a spatula, the robot must reason over the contact formation (e.g., point, line, or patch) to ensure the spatula is scraping properly at the desired region on the wok. In this section, we evaluate VIRDO++ ability to estimate extrinsic contact features; specifically, the contact line created as the spatula is brought into contact with the table. To this end, we compute the signed distance values of the tables surface ($z = 0$) to compute a contact cluster. This requires only a single feed-forward path where the input is an $N \times 3$ matrix, where N is the number of query points. From the contact cluster, we extract a contact line by selecting the two points furthest apart similar to [40, 41, 42]. The evaluation metrics are the average Euclidean distance $\frac{1}{2}\Sigma_{i=0}^{1}||l_i - l_i^*||$, where $l_i$ is the detected contact lines, $l_i^*$ is the ground truth contact lines, and $i = 0, 1$ refers to right and left ends of each contact line respectively. Ground truth contact lines were obtained from ground truth pointcloud without occlusions by applying a contact mask $z \leq \epsilon$. This contact feature estimation task is particularly interesting as it is the inverse problem of [1]; estimating [1]'s privileged input (i.e., contact location) by inferring $c_t$.

As shown in Tab. 2 and Fig. 5, VIRDO++ can accurately detect contact locations with less than 8 mm error on a flat table for the current and the following time step regardless of occlusions at the tip. As in Sec.4.2, test trajectories from 4 spatulas with 15% occlusion were used for the evaluation. Tab.2 shows that the average contact line detection errors from all 4 spatulas are less than 7 [mm] during estimation and are less than 8 [mm] for predictions in Euclidean error. Similar to the result in Sec. 4.2, estimations are 1 [mm] more precise than predictions because they are optimized using the sensor measurements. Considering that the total length of the spatulas are about 200 [mm], this

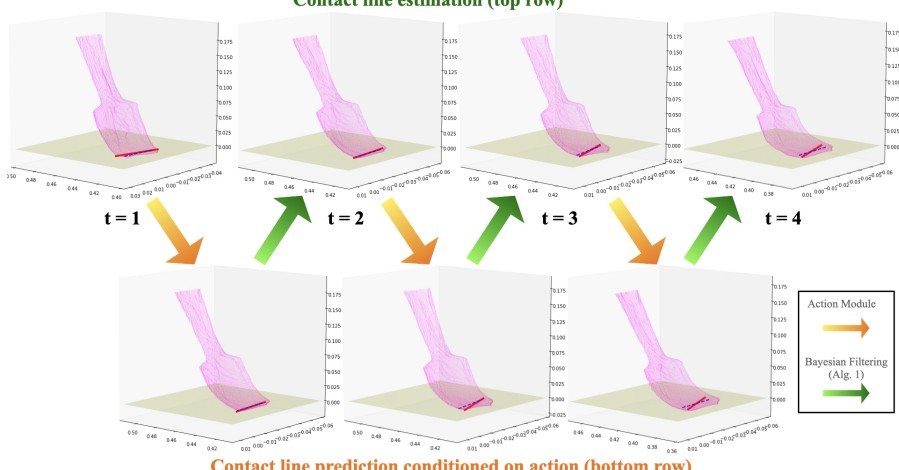

**Contact line estimation (top row)**

t = 1      t = 2      t = 3      t = 4

**Contact line prediction conditioned on action (bottom row)**

Fig. 5: VIRDO++ detects the line of contact throughout an alternating sequence of "estimation → prediction → estimation → ...". Bayesian filtering increase the accuracy of contact location detection as the red lines (detected contact line) gets closer to the dotted blue lines (ground truth) following the green arrow. In each figure, yellow plane: table location, magenta spatula: reconstructed geometry from VIRDO++ , red line: detected contact line, blue: ground truth contact line. All axis are in [m] scale in world coordinate.

| L2 Error [mm] | VIRDO++ | w/o $c_t$ | Rigid body asmp. |
|---|---|---|---|
| Contact Est. | 6.647 (4.806) | 9.872 (5.725) | 21.203 (7.031) |
| Contact Pred. | 7.198 (4.439) | 9.356 (5.978) | 22.572 (6.990) |

Table. 2: Having latent embedding $c_t$ as an input, VIRDO++ can decrease about 33 % of error for estimations and 23 % for predictions. Moreover, rigid body assumption is a worst case scenario producing ×3 error.

is less than 4% of error in total length. Estimating extrinsic contacts is a challenging task for rigid object [43] and is further complicated by object compliance; however, our approach proves effective due to its high-quality 3D reconstructions (details of error distribution in **Appendix** Fig. 12). Tab. 2 also quantifies performance against ablations (without $c_t$ and Rigid-body assumption) illustrating the importance of the contact embedding and reasoning over compliance.

## 4.4 Generalizations Tasks

This section demonstrates VIRDO++ 's generalization ability to unseen objects and their dynamics. For each unseen object, the generalization dataset contains 1 nominal point cloud and 3 trajectories, each with 24 transitions with occlusion. Here, we directly apply the same pre-trained model from Sec. 4.1 to the generalization test dataset.

### 4.4.1 Object code inference for Unseen Objects

VIRDO++ is able to represent nominal geometries unseen during training within distribution. Fig. 6 shows reconstructions of both seen and unseen object in Fig. 3 using a single pretrained Object Module. For the unseen objects, we start from a randomly initialized object code $\alpha \sim \mathcal{N}(0, 0.01)$ and update the code by gradient descent with Adam. We use the same loss function from Eq. 1 with pre-trained VIRDO++ while freezing the weights of the network. The first row of Tab. 3 shows the Chamfer distance error of the unseen objects alongside training objects, indicating the relatively small drop in performance. When comparing

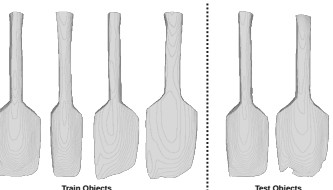

Fig. 6: VIRDO++ can reconstruct train and test objects with high fidelity (same order as Fig. 3).

Fig. 6 and the ground truth in Fig. 3, we note that the test reconstructions are close to the ground truth aside from some fine edge details.

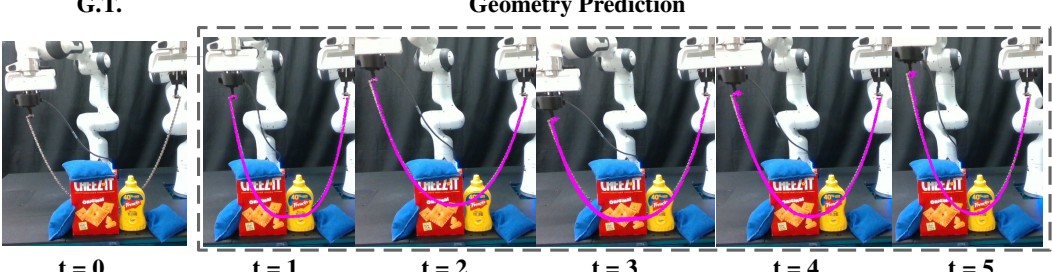

| G.T. | | Geometry Prediction | | | |
| t = 0 | t = 1 | t = 2 | t = 3 | t = 4 | t = 5 |

Fig. 7: VIRDO++ can accurately estimate the bike chain geometry (predictions overlaid in magenta) under occlusions due to objects at the front. Note that the chain intermittently makes contact and rests on the table, which is correctly predicted by VIRDO++ despite significant occlusion of where the chain makes contact.

### 4.4.2 Zero Shot Dynamics Prediction and State-estimation Generalization

With the object codes obtained in Sec. 4.4.1, VIRDO++ is able to predict the unseen object's dynamics based on their nominal shape. Here, we evaluate both prediction and state-estimation performance as in Sec. 4.2. We note that the observations of the unseen objects include occlusions as discussed in **Dataset**. Tab. 3 shows comparable performance between training and test objects with a small drop in accuracy, highlighting our model's generalization capability. Sec. A.3.6 in **Appendix** provides intuition for the Chamfer distance values. Further, we evaluate VIRDO++ 's ability to perform high-fidelity geometry prediction and estimation even in novel environment (Wok scraping) without 3D model. Details can be found in **Appendix** Sec. A.4.

| CD $(\times 10^3)$ | VIRDO++ | w/o $c_t$ | Rigid Asm. | Test Obj. 1 | Test Obj. 2 |
|---|---|---|---|---|---|
| Geo. – Nom. | 0.65 (0.08) | - | - | 1.08 | 1.29 |
| Geo. – S.E. | 0.93 (0.21) | 1.18 (0.35) | 16.66 (11.51) | 1.39 (0.20) | 1.56 (0.29) |
| Geo. – Pred. | 1.04 (0.35) | 1.09 (0.25) | 15.16 (10.57) | 1.54 (0.45) | 2.40 (1.08) |
| Wr. – Pred. | 1.14 (1.05) | 1.04 (1.01) | 1.63 (0.89) | 1.50 (1.00) | 2.14 (1.75) |

Table. 3: (Column 1-3) Ablation study using the train object's unseen trajectories. Numbers indicate mean (std.) of each error distribution. Chamfer distance used 5600 points and is multiplied by $10^3$. (Column 4-5) Generalization results to unseen objects using the same pretrained VIRDO++ . Geo. = Geometry, Nom. = Nominal, S.E. = State Estimation, Pred. = Prediction, Wr. = Wrench calculated as $\|\hat{\boldsymbol{f}}_t - \boldsymbol{f}_t\|$.

### 4.5 Bike Chain Application

In this experiment, the chain ends are grasped by two arms. One end of the chain is moved randomly through the vertices of an equally-spaced $6 \times 6 \times 6$ grid fitting inside of a $0.1 \times 0.18 \times 0.2$ meter bounding box. The visited vertices of this grid make up the training dataset. The test dataset is generated by sampling 20 random positions uniformly from the bounding box. The end-effector orientations were fixed; therefore, $\boldsymbol{p}_t, \boldsymbol{a}_t \in \mathbb{R}^3$. As shown in Fig. 7, VIRDO++ is able to predict/estimate both free-space and in-contact chain geometry. The in-contact phase occurs when the arm moves sufficiently far down such that the chain lays on the table. We report the quantitative results of geometry estimation and dynamics prediction in Tab. 1 and **Appendix** Sec. A.3.3. The empirical results are comparable to spatula experiments despite significantly larger shape variations. Details on wrench prediction errors in **Appendix** Sec. A.3.4.

## 5 Limitations

Our proposed approach has a number of limitations: **1) Non-rigid grasps:** VIRDO++ assumes that the gripper is rigidly holding the object. When it comes to non-rigid grasping, we need to estimate relative motion of the objects w.r.t the gripper and the force transmission is now a function of this relative motion. One potential solution is to use collocated tactile sensor feedback [44, 45, 46] with F/T sensing to reason over reaction forces conditioned on the grasp. **2) Inertial effects:** our approach assumes quasi-static manipulation (negligible accelerations). During fast motions, the tools' inertial effects may also impact deformations and wrench measurements. When applicable, we suggest supplementing VIRDO++ 's input with end-effector accelerations. **3) Generalization to novel environments:** While we provide an example of environment generalization (**Appendix** Sec. A.4), the performance is lower than our other demonstrated applications. VIRDO++ may benefit from a 3D model of the environment to integrate inter-penetration losses [23] or collision checkers [47].

## Acknowledgments

This work was in part supported by Robotics at Google under the Google Faculty Research Award 2021 and the National Science Foundation (NSF) grant NRI-2220876. Any opinions, findings, and conclusions or recommendations expressed in this material are those of the authors and do not necessarily reflect the views of the National Science Foundation.

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
