# OpenReview forum: "VIRDO++: Real-World, Visuo-tactile Dynamics and Perception of Deformable Objects"
_robot-learning.org/CoRL/2022/Conference — CoRL 2022 Poster_

### Official Review · Reviewer_NDvN · 2022-07-30

**Originality:** Fair
**Technical Quality:** Fair
**Clarity Of Presentation:** Poor
**Impact:** 2

**Recommendation:**

Strong Reject: I recommend rejecting the paper and will argue for my recommendation even if other reviewers hold a different opinion.

**Summary:**

The paper proposes a deformation dynamic model of deformable objects and its complementary model of geometry estimation using multi-modal visuo-tactile information,, which is demonstrated in two real world tasks, spatula manipulation and bike chain application.

**Issues:**

1. The section 4 on related work should be placed right after the introduction for the sake of understanding the relevant background.
2. In the paper, the authors should describe models or methods with technique details instead of using meaningless words conveying limited information. For example, in the abstract, the phrase "a new formulation for deformation dynamics and a complementary state-estimation algorithm that (i) maintains a belief over deformations, and (ii) enables practical real-world application by removing the need for privileged contact information."  fails to tell the key contributions of the paper with strong supporting details.
3. Some of the figures and tables should be formatted more properly. For example, Fig.2, 3 and Table 2.
4. The conclusion on the generalization capability of the proposed model in the spatula manipulation task is not well supported by the results in Table 2a and Fig. 7. Some of the measuring scores double for the test object 2 compared to the training object.

**Quality Of The Limitations Section:**

Limitations are addressed clearly

**Reviewer Expertise:**

3: The reviewer is fairly confident that the evaluation is correct

**Robotics Focus:**

Relevant but unlikely to deploy to hardware in near future

**Strengths And Weaknesses:**

The main weakness of the paper is the presentation of the paper is not well organized and clear and the wording of the paper makes the contents difficult to understand. The paper needs to be improved significantly to convey the search findings more effectively.

**Summary Of Recommendation:**

The presentation of the paper is not well organized and clear and the wording of the paper makes the contents difficult to understand. Hence I recommend rejecting the paper.

---

> ### Author Response · Authors · 2022-08-28
> **Response to Reviewer NDvN**
>
> **Comment:**
>
> We want to thank you for your time and diligence in reading our paper and providing feedback, we are grateful. We have worked hard to address the issues and weaknesses you have raised in the updated manuscript and supplementary materials  (highlighted in blue). Below we discuss these additions and edits:
>
> **1. The section 4 on related work should be placed right after the introduction for the sake of understanding the relevant background.**
>
> Thank you for the note, the related work is now directly after the introduction.
>
> **2. In the paper, the authors should describe models or methods with technique details instead of using meaningless words conveying limited information.**
>
> We’ve made several modifications to the manuscript to improve its clarity. For instance, the particular example mentioned in abstract has now been updated to “a new formulation for deformation dynamics and a complementary state-estimation algorithm that (i) maintains a belief distribution of deformation within a trajectory, and (ii) enables practical real-world application by removing the need for contact patches.” We’ve also included references for some of the terms that may not be entirely clear. For example, we have provided an explicit definition for the contact embedding in Section 3.1 (Line 91-92). This definition was part of the original manuscript and may have been missed by the reviewer pointing this out.
>
> **3. Some of the figures and tables should be formatted more properly. For example, Fig.2, 3 and Table 2.**
>
> Thank you for the comment. We’ve made a significant effort in reformatting the Figures and Tables. For instance, **Figure 2** (Page 3) is now a full text width single row and **Figure 3** (Page 5) benefits from better placement. **Table 2** (Page 8) is now standalone for better clarity and we’ve introduced an additional table to declutter the presentation. **Figure 9** (Page 9) is now also corrected for better axes labels.
>
> **4. The conclusion on the generalization capability of the proposed model in the spatula manipulation task is not well supported by the results in Table 2a and Fig. 7. Some of the measuring scores double for the test object 2 compared to the training object.**
>
> We want to emphasize that the Chamfer Distance numbers reported are multiplied by 1,000. A doubling of the score is, from a practical point of view, very small. For example, consider the Rigid body assumption and note that the performance worsens by more than an order of magnitude. This puts the nearly doubling value into perspective.
>
> **Zip File:**
>
> /attachment/2ecd2d83e7b2a5a18e045061a992be54848a6532.zip

---

### Official Review · Reviewer_8Dpk · 2022-07-31

**Originality:** Fair
**Technical Quality:** Poor
**Clarity Of Presentation:** Good
**Impact:** 3

**Recommendation:**

Weak Reject: I recommend rejecting the paper, but will not argue for my recommendation if the majority of other reviewers have a different opinion.

**Summary:**

This work builds on recent progress in multimodal neural implicit representations for deformable object manipulation by proposing (1) an action-conditioned dynamics model to predict future deformations and (2) a state estimation algorithm to maintain a belief distribution over object deformations. The work performs two real-world tasks to demonstrate generalization to unseen objects and novel environments.

**Issues:**

For their current work, I'd suggest adding more experiments with newer geometries (eg: other objects besides spatula, bike chain), thorough interpolation to intermediate sizes / shapes (at least 3-5).

This work is highly incremental, and I am not convinced with the technical novelty of the work, so would like to see if authors considered any other approaches to model architecture itself.


**Quality Of The Limitations Section:**

Additional details required

**Reviewer Expertise:**

3: The reviewer is fairly confident that the evaluation is correct

**Robotics Focus:**

Sufficient demonstration on hardware

**Strengths And Weaknesses:**

strengths :
(1) the work enables real world application of deformable manipulation.
(2) training, loss formulations, and algorithms are clearly presented

weaknesses :
(1) the work lacks originality, it heavily builds upon the predecessor work (VIDRO). It relies on the original VIDRO architecture and replaces (a) contact patches with contact embedding and (b) adding an additional dynamics model (to focus on non-static scenes).
(2) real-world experiments aren't fully convincing. The paper shows interpolation on limited geometries which is insufficient to draw conclusions.

**Summary Of Recommendation:**

I read the paper thoroughly and used my best judgement in assessing the paper.

---

> ### Author Response · Authors · 2022-08-28
> **Response to Reviewer 8Dpk**
>
> **Comment:**
>
> We want to thank you very much for your time and diligence in reading our paper and providing detailed feedback, we are grateful. We have worked hard to address the issues and weaknesses you have raised in the manuscript and supplementary materials (highlighted in blue). Below we discuss these additions and edits:
>
> **1. It relies on the original VIDRO architecture and replaces (a) contact patches with contact embedding and (b) adding an additional dynamics model (to focus on non-static scenes). I am not convinced with the technical novelty of the work, so would like to see if authors considered any other approaches to model architecture itself.**
>
> To address this concern of model architecture and baselines, we have added a new section to the **Appendix A.4** that considers two types of comparisons to other approaches (**Table 7**). First, we compare against an explicit geometric representation (GRNet) which has shown state-of-the-art performance in point cloud processing (reconstruction and shape completion) for 3D geometries. Second, we study implicit methods which are ablations of the VIRDO architecture using DeepSDF and SIREN for 3D geometry representations. These other approaches are evaluated along with the VIRDO architecture for their efficacy in representation and prediction. All approaches are evaluated on real-world data and we show that the VIRDO architecture outperforms the other methods.
>
> $$\\begin{array} {rrrrr} \hline
> CD (\times 10^3) &  \text{VIRDO}^{++}  & \text{DeepSDF} & \text{SIREN} & \text{Pos. Enc.} \\\\ \hline
>   \text{Train Est.}  & 0.777 (0.215)   &  5.087 (0.737) & 5.521  (1.015) & 5.569 (2.847) \\\\ \hline
>         \text{Train Pred.}  &  0.830 (0.330)  &  5.667 (1.306) &5.457 (0.888) &  5.886 (2.561) \\\\ \hline
>         \text{Test Est.}  & 0.934 (0.210)  &  10.016 (3.253) & 6.349 (1.388) & 8.081 (3.790) \\\\ \hline
>         \text{Test Pred.} & 1.041 (0.348) & 9.552 (2.767) & 6.988 (1.280) & 7.188 (3.603) \\\\ \hline
> \end{array}$$
>
> In addition to the above, we’d like to draw your attention to the other contribution of the manuscript, an accompanying Bayesian inference algorithm that allows the robot to maintain a belief distribution over the deformed geometries of objects. Our proposed method builds on the particle filter to iteratively refine the probability distribution. Further, we demonstrate variations of the algorithm in the supplementary materials (new addition) and show performances on real-world data.
>
>
> Finally, we’d like to emphasize the importance of extensions to the architecture and learning approach to transition VIRDO into the real-world. Without the methods and approaches proposed in our work, it would not be possible to deploy VIRDO in the real-world and this represents a significant advance forward. Real-world deformable objects we used are more complex than the dynamics of homogeneous objects such as dolls/cloth/rope/dough. Spatulas we used are heterogeneous (wood, plastic, silicone) as the silicon tips are 'wrapping' the wood/plastic sticks, and the bike chain consists of dozens of links.
>
> **2. Real-world experiments aren't fully convincing. The paper shows interpolation on limited geometries which is insufficient to draw conclusions.**
>
>
> Thank you for the feedback, we’d like to clarify that our primary axis of generalization is not to new shapes, but to new deformations. This is a subtle but important difference. To help with the clarification of this important difference, we’ve added a new **Section A.3.1** to the supplementary materials. Here, we illustrate the large range of deformations that our training and test spatula undergo while they are brought into contact in the same configuration with the same force by the robot. The key observation is that this variety in spatula deformations is due to their variance in stiffness – parameters that are not accessible to our approach. Generalization toward unseen contact formations is therefore a critical functionality that our approach delivers.
>
>
>
>
>
> **Zip File:**
>
> /attachment/469287119faef5b058df12d81dbb94a6d25a9360.zip

---

### Official Review · Reviewer_fX59 · 2022-08-01

**Originality:** Good
**Technical Quality:** Very Good
**Clarity Of Presentation:** Excellent
**Impact:** 4

**Recommendation:**

Weak Accept: I recommend accepting the paper, but will not argue for my recommendation if the majority of other reviewers have a different opinion.

**Summary:**

The paper presents a visuo-tactile state estimation and dynamics model for deformable objects, built on the previous work VIRDO. This work improves VIRDO in two aspects: (1) it removes the need for priviledged information (contact patch) by estimating a contact embedding, which enables real world robotic application; (2) it learns a implicit dynamics model. Based on the this multimodal neural implicit representation, the authors introduce a state estimation algorithm by maintaining a belief distribution over object deformations.

**Issues:**

* The plots in Fig.5 should be annotated with the names of the quantities. For example, the first one should be "Geometry prediction and state estimation", and second one should be "force and toque prediction".

**Quality Of The Limitations Section:**

Limitations are addressed clearly

**Reviewer Expertise:**

4: The reviewer is confident but not absolutely certain that the evaluation is correct

**Robotics Focus:**

Sufficient demonstration on hardware

**Strengths And Weaknesses:**

Strengths:
* It addresses the biggest limitation of VIRDO - cannot be directly applied to real-world robotics settings due to the need for priviledged information, which is a significant contribution.
* It integrates an implicit dynamics model with VIRDO seamlessly, and show the advantages of proposed visuo-tactile representation with robotic application (state estimation), in real-world. The results look great and seem to be robust to occlusion, thanks to tactile information.

Weaknesses:
* The paper proposes an implicit dynamics model for deformable object, but doesn't compare it with an explicit counterpart. For example, one may extract a mesh from predicted SDF and learn an explicit dynamics model on the mesh.
* Some of the design choices are not ablated. For example, why do you choose to use hypernetwork for deformation and object module? One alternative is to concatenate query point with the feature and feed them to a MLP (like DeepSDF)
* It's not evaluated on downstream manipulation tasks. I'm not so sure about how to interpret the state estimation results. It will be nice if the author can evaluate the proposed method on a manipulation task, to give us a sense of how accurate the state estimation and dynamics model need to be to, for example, flip a steak with a spatula.

**Summary Of Recommendation:**

The paper proposes a visuo-tactile representation for deformable object that can be applied to real world. The proposed approach is not groundbreaking, but well executed. I believe this will be interesting for the wider robot learning community, especially deformable object manipulation. However, I have some concerns regarding the experiment section. I would be happy to update my rating if more results can be shown.

---

> ### Author Response · Authors · 2022-08-28
> **Response to Reviewer fX59**
>
> **Comment:**
>
> We want to thank you very much for your time and diligence in reading our paper and providing such detailed feedback. We particularly appreciate that you noticed the importance of the necessary contributions to bring VIRDO into the real world. We are excited that these modifications enabled accurate state estimation on real-world objects that are robust to occlusions. Based on your feedback and comments, we’ve made a number of major edits and additions to the manuscript (highlighted in blue) that we believe has significantly improved the quality of the work.
>
> Regarding the specific weaknesses you mentioned:
>
> **The paper proposes an implicit dynamics model for deformable objects, but doesn't compare it with an explicit counterpart. For example, one may extract a mesh from predicted SDF and learn an explicit dynamics model on the mesh.**
>
> Thank you for pointing out this shortcoming. In order to address this, we have added a new section to the **Appendix A.4.2** (Line 530-555) that considers two types of comparisons to other approaches summarized in **Table 8** (Page 17). First, we compare against an explicit representation (GRNet) which has shown state-of-the-art performance in point cloud processing (shape completion) for 3D geometries. We also include implicit methods which are ablations of the VIRDO architecture using DeepSDF and SIREN for 3D geometry representations.
>
> **All approaches are evaluated in the real-world and show that the proposed approach outperforms them.
> Some of the design choices are not ablated. ... One alternative is to concatenate query point with the feature and feed them to a MLP (like DeepSDF)**
>
> Thank you for pointing this out, as mentioned in response to your previous observation, we have conducted a study evaluating ablations of our method (DeepSDF, SIREN, and positional encoder specifically) and have summarized their results in **Table 8** (Page 17). We find that VIRDO++ base architecture works better than these approaches.
>
> $$\\begin{array} {rrrrrr} \hline
> CD (\times 10^3) &  \text{VIRDO}^{++}  & \text{Explicit Dyn.} & \text{DeepSDF} & \text{SIREN} & \text{Pos. Enc.} \\\\ \hline
>   \text{Train Est.}  & 0.777 (0.215)  &  - &  5.087 (0.737) & 5.521  (1.015) & 5.569 (2.847) \\\\ \hline
>         \text{Train Pred.}  &  0.830 (0.330) & 0.759 (0.667) &  5.667 (1.306) &5.457 (0.888) &  5.886 (2.561) \\\\ \hline
>         \text{Test Est.}  & 0.934 (0.210)  & -  &  10.016 (3.253) & 6.349 (1.388) & 8.081 (3.790) \\\\ \hline
>         \text{Test Pred.} & 1.041 (0.348) &  1.230 (0.826) & 9.552 (2.767) & 6.988 (1.280) & 7.188 (3.603) \\\\ \hline
> \end{array}$$
>
>
>
>
> **It's not evaluated on downstream manipulation tasks. I'm not so sure about how to interpret the state estimation results.**
>
>
> Thank you for the recommendation! To address this, we have added a downstream manipulation task – estimating and predicting the extrinsic contact made between the object and the environment in **Section 4.3** (Page 7-8). The key idea is to estimate and predict the contact line created between the tool and environment as the tool is brought into contact with the environment. We can achieve this task because our method is able to produce high-fidelity full 3D shape estimations to a level of accuracy sufficient to detect contact between two objects. The details of this task are provided in the manuscript Line 221-253. We leave more complex tasks such as flipping a steak to future work as there was not enough time during the rebuttal phase.
>
> $$\\begin{array} {rrrr} \hline
>         \text{L2 Error} [mm] & \text{VIRDO}^{++} & \text{Without} ~ c_t& \text{Rigid body asmp.} \\\\ \hline
>         \text{Contact Est.}  & 6.647 (4.806) &  9.872 (5.725) & 21.203 (7.031) \\\\
>         \text{Contact Pred. } & 7.198 (4.439) & 9.356 (5.978)  &  22.572 (6.990)\\\\ \hline
> \end{array}$$
>
>
> **The plots in Fig.5 should be annotated with the names of the quantities. For example, the first one should be "Geometry prediction and state estimation", and second one should be "force and torque prediction".**
>
>
> Thank you for pointing this out, we’ve corrected **Fig. 5** in Page 6 with new titles and axis labels.
>
>
> **Zip File:**
>
> /attachment/53f7c17c188794fb89f2b61f9a8cc47d2cafddea.zip

---

> > ### Author Response · Authors · 2022-08-28
> > **Response to Reviewer fX59**
> >
> > In summary, we’ve added an explicit method and ablations of our method as benchmarks and also proposed an example downstream task (extrinsic contact estimation). We believe that these edits and modifications have significantly improved the quality of the manuscript and are grateful for your feedback. We hope that you’ll consider updating your rating and look forward to any additional comments you may have!
> >
> > At our kindest regards ,

---

### Official Review · Reviewer_y9NZ · 2022-08-07

**Originality:** Good
**Technical Quality:** Very Good
**Clarity Of Presentation:** Very Good
**Impact:** 3

**Recommendation:**

Weak Accept: I recommend accepting the paper, but will not argue for my recommendation if the majority of other reviewers have a different opinion.

**Summary:**

This paper builds on top of the prior work on VIRDO and proposes a framework for predicting deformation of deformable objects conditioned on the multi-modal observation such as partial point cloud and force beedback. Compared to prior work of VIRDO, the new appraoch removes the assumption on known contact patches, learns the deformation dynamics and develops a gradient-based state estimation method able to handle occlusion. The proposed method is demonstrated on real manipulation trajectories.

**Issues:**

Please see above.

**Quality Of The Limitations Section:**

Limitations are addressed clearly

**Reviewer Expertise:**

4: The reviewer is confident but not absolutely certain that the evaluation is correct

**Robotics Focus:**

Sufficient demonstration on hardware

**Strengths And Weaknesses:**

### Strength
1. The paper makes a clear contribution compared to prior works on VIRDO. I appreciate the added component of the deformation dynamics model, which is essentially a latent dynamics model that avoides the need of known contact information.
2. The method is sufficiently demonstrated on real-world trajectories, showing the effectiveness of state estimation and dynamics prediction.
3. The paper is well presented and the supplementary video is very helpful.

### Weakness
I think the paper is a clear accept and the below weaknessnes are mainly suggestions to further improve the paper.

1. While the paper claims that the contect embedding c predicts the contact, it is actually not directly supervised the same as the prediction of reaction wrench. The contact embedding is only trained implicitly through other losses. As mentioned in the paper, contact embedding may also contain other information like the stiffness. It would be better to try reading out the contact information from c, or change the name of this component to something like a latent vector.

2. The paper can make a better connection to prior works. There are prior works on dealing with occlusion for other deformable objects such as cloth [1,2,3], where self-occlusion can be an important issue. The current description in line 248-250 can be misleading.

3. Does the method require full point cloud information during training? What forms of data are collected for training in the real world? Is there a multi-camera setup in place?

4. Finally, in the beginning I thought the dynamics model is used for planning in the two real world tasks demonstrated but in fact, they are only prediction experiments. It would be better to clarify this in the paper.

[1] Garmentnets:Category-level pose estimation for garments via canonical space shape completion

[2] Mesh-based Dynamics with Occlusion Reasoning for Cloth Manipulation

[3] Learning visible connectivity dynamics for cloth smoothing




**Summary Of Recommendation:**

Overall, the paper proposes an interesting method that overcomes some limitations of a prior work. The paper is well presetned with convincing experiments and thus I recommend acceptance.

---

> ### Author Response · Authors · 2022-08-28
> **Response to Reviewer y9NZ**
>
> **Comment:**
>
> We want to thank you very much for your time and diligence in reading our paper and providing such detailed feedback. We thank you for recognizing our contributions to bring VIRDO to real-world work by removing the need for knowledge of contact patches and extending it to a dynamics model. Also, we are happy to learn that you found the real-world tasks compelling and that the video was very helpful.
> Based on your feedback and comments, we’ve made a number of major edits and additions to the manuscript (highlighted in blue) that we believe has significantly improved the quality of the work.
>
>
> Regarding the specific weaknesses you mentioned:
>
> **1. While the paper claims that the contact embedding c predicts the contact, it is actually not directly supervised the same as the prediction of reaction wrench. It would be better to try reading out the contact information from c, or change the name of this component to something like a latent vector**
>
> Thank you for the comment, we decided that the latter option (changing the name) is more effective for clarity. We have changed the name from ‘contact embedding’ to ‘contact latent vector’ as you can find in Line 47, 93, 102, 142, 145, 148, and 212.
>
>
> **2. The paper can make a better connection to prior works. There are prior works on dealing with occlusion for other deformable objects such as cloth [1,2,3], where self-occlusion can be an important issue. The current description in line 248-250 can be misleading.**
>
>
> Thank you for the note and great references. We have included these references and a brief description into the manuscript, strengthening the connection to prior work (Line 69-72). We have copied the revision here: “These approaches have been used largely for 1D and 2D objects (rope, cloth, top view of cylindrical objects). Among them, structured representations (e.g. Mesh or GNN) are advantageous at tracking connectivity, mainly used for deformable objects with large self-occlusions (e.g. folded cloth or knot)”. Thank you again for the suggestion.
>
>
> **3. Does the method require full point cloud information during training? What forms of data are collected for training in the real world? Is there a multi-camera setup in place?**
>
>
> The method only requires a full point cloud from the nominal geometry of the object. This is collected by holding the object in front of the camera and rotating. Deformed object geometries are collected from the same 1 camera and are all partials (with potential front occlusions due to obstacles). A multi-camera setup may improve results though we found the approach works well without it. This is now clarified in **Section 4.1** (Line 176-181).
>
> **4. Finally, in the beginning I thought the dynamics model is used for planning in the two real world tasks demonstrated but in fact, they are only prediction experiments. It would be better to clarify this in the paper.**
> Thank you for pointing this out, we’ve added text to clarify that indeed the predictions are evaluated but not used for planning. The specific change was to ​​update the sentence in **Section 3.1** (Line 105-108) "This predictive capability enables state-estimation and planning for deformable object manipulation." $\to$ "This predictive capability enables state-estimation and downstream tasks such as contact location detection crucial for deformable object manipulation. Though we did not include it in this paper, the predictive ability of VIRDO++ also enables planning for deformable object manipulation."
>
>
> **Zip File:**
>
> /attachment/1847d58c9336395841de2832652d76d2486e2d24.zip

---

> > ### Author Response · Authors · 2022-08-28
> > **Response to Reviewer y9NZ**
> >
> > We are grateful for your feedback as it helped us significantly improve the quality of the manuscript. We made major updates, including 1) a comparison in the appendix **Section A.4.2** where we perform a benchmark against other approaches (explicit and implicit) on real-world data, and 2) an additional downstream manipulation task in **Section 4.3** of the manuscript where we estimate extrinsic contact lines. Please feel free to reach out if you have any further questions or comments!
> >
> >
> > At our kindest regards ,

---

### Meta-Review · Area_Chair_qy2A · 2022-09-07

**Recommendation:** Accept (Poster)
**Confidence:** 4

**Metareview:**

The authors propose a method to predict the reaction wrenches at the wrist of a robot arm and contact embeddings of deformable objects in manipulation tasks. The proposed approach builds on prior work, where most importantly, the need for privileged information was removed.

Strength:

    -Exploiting force measurements at the wrist for predicting object deformations is a relevant and important topic.
    -Compared to the prior work, the proposed extension enables real-world applications.
    -This feature is exploited in two impressive real-world experiments.

Weakness:

    -It is an incremental approach.
    -No comparison to other approaches, not even in simulations.
    -The selected real-world manipulation experiments do not demonstrate the full benefit or the limitations of the proposed approach. How would the learned model work on other tasks as shown in Fig. 8 using the wok.
    -The description is often vague, e.g., the terms 'privileged information' or 'contact embedding' should be defined explicitly.
    -The description can be improved, e.g., Fig 5 has no axis labels, KDE is not defined, and what is x1e3? What is the value of the contact embedding?

Update: The authors added more explanation and additional experiments which greatly improves the paper.

**Best Paper Nomination:**

No